# Impact of Amyloid Pathology in Mild Cognitive Impairment Subjects: The Longitudinal Cognition and Surface Morphometry Data

**DOI:** 10.3390/ijms232314635

**Published:** 2022-11-23

**Authors:** Hsin-I Chang, Shih-Wei Hsu, Zih-Kai Kao, Chen-Chang Lee, Shu-Hua Huang, Ching-Heng Lin, Mu-N Liu, Chiung-Chih Chang

**Affiliations:** 1Department of Neurology, Cognition and Aging Center, Institute for Translational Research in Biomedicine, Kaohsiung Chang Gung Memorial Hospital, Chang Gung University College of Medicine, Kaohsiung 833, Taiwan; 2Department of Radiology, Kaohsiung Chang Gung Memorial Hospital, Chang Gung University College of Medicine, Kaohsiung 833, Taiwan; 3Institute of Biophotonics, National Yang Ming Chiao Tung University, Taipei 112, Taiwan; 4Department of Nuclear Medicine, Kaohsiung Chang Gung Memorial Hospital, Chang Gung University College of Medicine, Kaohsiung 833, Taiwan; 5Center for Artificial Intelligence in Medicine, Chang Gung Memorial Hospital, Taoyuan 333, Taiwan; 6Bachelor Program in Artificial Intelligence, Chang Gung University, Taoyuan 333, Taiwan; 7Department of Psychiatry, Taipei Veterans General Hospital, Taipei 112, Taiwan; 8School of Medicine, National Yang Ming Chiao Tung University, Taipei 112, Taiwan

**Keywords:** amyloid, mild cognitive impairment, longitudinal, cognition, surface-based morphometry

## Abstract

The amyloid framework forms the central medical theory related to Alzheimer disease (AD), and the in vivo demonstration of amyloid positivity is essential for diagnosing AD. On the basis of a longitudinal cohort design, the study investigated clinical progressive patterns by obtaining cognitive and structural measurements from a group of patients with amnestic mild cognitive impairment (MCI); the measurements were classified by the positivity (Aβ+) or absence (Aβ−) of the amyloid biomarker. We enrolled 185 patients (64 controls, 121 patients with MCI). The patients with MCI were classified into two groups on the basis of their [^18^F]flubetaben or [^18^F]florbetapir amyloid positron-emission tomography scan (Aβ+ vs. Aβ−, 67 vs. 54 patients) results. Data from annual cognitive measurements and three-dimensional T1 magnetic resonance imaging scans were used for between-group comparisons. To obtain longitudinal cognitive test scores, generalized estimating equations were applied. A linear mixed effects model was used to compare the time effect of cortical thickness degeneration. The cognitive decline trajectory of the Aβ+ group was obvious, whereas the Aβ− and control groups did not exhibit a noticeable decline over time. The group effects of cortical thickness indicated decreased entorhinal cortex in the Aβ+ group and supramarginal gyrus in the Aβ− group. The topology of neurodegeneration in the Aβ− group was emphasized in posterior cortical regions. A comparison of the changes in the Aβ+ and Aβ− groups over time revealed a higher rate of cortical thickness decline in the Aβ+ group than in the Aβ− group in the default mode network. The Aβ+ and Aβ− groups experienced different *APOE* ε4 effects. For cortical–cognitive correlations, the regions associated with cognitive decline in the Aβ+ group were mainly localized in the perisylvian and anterior cingulate regions. By contrast, the degenerative topography of Aβ− MCI was scattered. The memory learning curves, cognitive decline patterns, and cortical degeneration topographies of the two MCI groups were revealed to be different, suggesting a difference in pathophysiology. Longitudinal analysis may help to differentiate between these two MCI groups if biomarker access is unavailable in clinical settings.

## 1. Introduction

Mild cognitive impairment (MCI) represents a state between normal cognition and early dementia [1]. Although MCI is a general construct and is not necessarily progressive or the earliest stage of Alzheimer’s disease (AD), it is most often studied in this context and is commonly regarded as the earliest clinical manifestation of AD pathophysiology [2]. The National Institute on Aging and the Alzheimer’s Association (NIA-AA) published clinical criteria for MCI due to AD, and these criteria suggest that the cognitive capacity of a patient with MCI due to AD is less than the level corresponding to their age, gender, and education but not at the level of dementia [2]. The NIA-AA research criteria was further expanded to include pathological biomarkers for defining MCI due to AD [2]. A growing body of evidence suggests that patients with MCI due to AD who exhibit a positive amyloid status are more likely than those without this status to further progress to dementia [3,4]. Direct comparisons between patients with MCI with (Aβ+) or without (Aβ−) amyloid burden can help to delineate the mechanistic role of amyloid in neurodegeneration. In a clinical setting, constructing a group-based neuroimaging model based on clinical and magnetic resonance imaging (MRI) data is rational because such data are accessible and closely correlated to clinical features [5,6].

Neurodegeneration is defined as the age-associated deterioration of neuronal function that leads to cognitive decline [7,8,9]. Various MRI processing toolboxes have been developed to determine structural changes in degenerative disorders. The parcel morphometry matrix, which considers measurements such as cortical thickness or gray matter volume, can be used to identify patients with MCI [10,11] or AD [12,13]. Longitudinal cortical thickness measurement may be helpful for modeling neurodegeneration trajectory [14]. Current large-scale longitudinal neuroimaging studies on MCI due to AD have emphasized individual changes in neuroimaging measurements over time, and the effects of the predictors of interest (e.g., diagnostic group or interactions between time and group) can help clarify individual changes that occur.

We hypothesized that longitudinal cognition and surface morphometric changes could help differentiate between Aβ+ and Aβ− MCI groups. Direct comparisons of these two groups can help delineate degeneration patterns by amyloid burden. The differentiation of these two groups is crucial to the development and implementation of treatment strategies. Whether the effects of apolipoprotein E4 (*APOE* ε4) on the rates of clinical and structural decline are consistent between these two groups is a topic that warrants further exploration. In the present study, we analyzed baseline and longitudinal cognitive data obtained from older adults without cognitive impairment, older adults with MCI due to AD and a positive amyloid biomarker status, and older adults with MCI due to AD and a negative amyloid biomarker status. We investigated how group differences, time effects, and group–time effect interactions influenced the cognitive and structural changes in these older adults. A linear mixed-effects (LME) model based on cortical thickness combined with baseline and follow-up cognitive measurements was used to assess the amyloid status of older adults with MCI due to AD.

## 2. Results

### 2.1. Baseline Demographics and Cognitive Tests

The control and experimental groups did not exhibit differences in gender, educational level, or the distribution of diabetes, hypertension, and hyperlipidemia (Table 1). The mean number of follow-up months was 40 months; the shortest and longest durations were 13 and 187 months, respectively. The Aβ+ group had a significantly higher proportion of *APOE* ε4 carriers relative to the Aβ− and control groups, which had a similar proportion of *APOE* ε4 carriers.

At baseline, the mini-mental state examination (MMSE) and cognitive abilities screening instrument (CASI) scores of the Aβ+ and Aβ− MCI groups were significantly lower than those of the controls (Table 1); relative to the controls, both the Aβ+ and Aβ− MCI groups scored lower in the CASI subdomains of short-term memory (STM) and orientation and verbal fluency; however, the STM scores of the Aβ+ MCI group were significantly lower than those of the Aβ− MCI group. Regarding Chinese-version verbal learning test (CVLT) results at baseline, the Aβ+ group obtained significantly lower scores in the learning trials (T1–T4), 30-s and 10-min recall tests, and cue-correct and cue-recall test. Relative to the controls, the salient memory deficits of the Aβ− MCI group were lower for the 30-s recall test, 10-min recall test, and cue-recall test. The learning trial scores and cue-correct scores of the Aβ− MCI and control groups were not significantly different.

We assessed the effect of gender on the cognitive measurements of the two MCI groups and discovered that gender did not have an effect on such measurements in the Aβ+ group (total CASI [β = −3.972, *p* = 0.47]; STM [β = −2.564, *p* = 0.47)) and Aβ− group (total CASI [β = 0.037, *p* = 0.987]; STM (β = −0.225, *p* = 0.739)).

### 2.2. Differences in Cortical Thickness in the Control and Experimental Groups at Baseline

The differences between the groups regarding cortical thickness (adjusted for age, gender, and educational level) and estimated total intracranial volume (eTIV) are presented in Figure 1A. Compared with the controls, the Aβ+ MCI group exhibited thinner cortical thickness, mainly in the entorhinal cortex and several scattered cortical regions. Differences between the Aβ− MCI and control groups were observed with respect to the medial prefrontal and temporal–parietal regions. Differences in cortical thickness (primarily the entorhinal area) were detected between the Aβ+ and Aβ− MCI groups.

### 2.3. Longitudinal Analysis: Cognitive Decline (MMSE and CASI)

Both the Aβ+ and Aβ− MCI groups obtained lower MMSE (Appendix A) and total CASI scores (Appendix A) relative to the controls. This decrease in MMSE and CASI scores was related to disease duration in the Aβ+ MCI group but not in the Aβ− MCI group. We further analyzed the scores from baseline to the 5-year follow-up point by using polynomial second-order trendlines (with 95% confidence intervals, Figure 2). On the basis of the memory performance differences between the Aβ+ and Aβ− groups at baseline, we further analyzed their STM scores (Appendix A), CVVLT delay recall (10 min), and cue-correct scores (Appendix A). Aβ+ MCI was revealed to have a significant group effect and group × disease duration interaction effects on all three scores; Aβ− MCI was revealed to have a significant group effect on STM score and a significant interaction effect on cue-correct score (Table 2).

In addition to STM, the eight cognitive domains of the CASI were segregated into executive domains (Appendix A) and nonexecutive domains (Appendix A). Aβ+ and Aβ− MCI did not have a group effect on the executive domains but significantly influenced two nonexecutive domains (i.e., the long-term memory and orientation domains). Aβ+ × disease duration influenced all eight domains, whereas Aβ− disease duration only influenced the orientation subdomain (Appendix A, Table 2). These findings are presented as Appendix A.

We assessed the effect of gender on longitudinal cognitive decline in the two MCI groups and discovered that gender did not influence the results of the Aβ+ group (total CASI (β = −6.064, *p* = 0.138, STM β = −0.323, *p* = 0.75); male gender as a reference) and Aβ− group (CASI (β = −3.755, *p* = 0.279, STM β = −0.810, *p* = 0.313); male gender as a reference).

### 2.4. Longitudinal Analysis: Cortical Thickness of Diagnostic Groups over Time

The cortical thickness neurodegeneration associated with Aβ+ MCI (Figure 1B) exhibited a gradient pattern that emphasized the medial–anterior lateral temporal areas. The annual changes in Aβ+ MCI (Figure 1C) suggested that cortical thinning started in the medial temporal, precuneus, and lateral temporal regions. In the Aβ− MCI group (Figure 1B), neurodegenerative clusters were scattered in the cortex regions, with such clusters noted in the lateral temporal and occipital areas, precuneus area, fusiform area, and inferior temporal area. For annual changes (Figure 1C), cortical thinning started in the temporoparietal and medial occipital regions in the Aβ− MCI group. In the control group, no significant neurodegeneration pattern was detected.

### 2.5. Longitudinal Analysis: Degenerative Pattern Differences between the Three Groups (Figure 3A)

Relative to the controls (Figure 3A1), the Aβ+ MCI group exhibited significant changes in cortical thickness over time, and these changes occurred diffusely in the temporal pole, entorhinal cortex, temporal, parietal, and frontal regions. Compared with the controls, the Aβ− group (Figure 3A2) exhibited more atrophy in the posterior and lateral temporal cortical regions. Relative to the Aβ− MCI group, the Aβ+ MCI group experienced changes in the cortical thickness in the entorhinal and anterior temporal poles at a faster rate (Figure 3A3).

### 2.6. Main and Time Effects of APOE ε4 on Cortical Thickness

In the Aβ+ MCI group, *APOE* ε4 carriers exhibited more scattered cortical atrophy relative to noncarriers (Figure 3B, decreased cortical thickness in *APOE* ε4 carriers with Aβ+ status). However, the effect of *APOE* ε4 on the degenerative process was extensively spread (Figure 3C, *APOE* ε4 thickness over time in Aβ+ group); specifically, the effect was more pronounced among *APOE* ε4 carriers than among the overall Aβ+ sample (Figure 1B, time effect of Aβ+ status). In the Aβ− MCI group, *APOE* ε4 carriers experienced a greater decrease in cortical thickness within the default mode network (DMN) relative to noncarriers (Figure 3B, decreased cortical thickness in *APOE* ε4 carriers with Aβ−). Furthermore, a longitudinal analysis revealed the significant genetic effects of *APOE* ε4 on disease progression (Figure 3C, *APOE* ε4 thickness over time in Aβ− group). The topographic distribution of cortical atrophy × time interactions in *APOE* ε4 carriers with Aβ− MCI mostly corresponded to that of the overall Aβ− MCI sample (Figure 1B, Aβ− time effect). The effects of *APOE* ε4 were not detected in the controls.

### 2.7. Cortical–Cognitive Relationships in Aβ+ and Aβ− MCI Groups

At baseline, MMSE, STM, orientation, and verbal fluency scores were significantly lower in the Aβ+ and Aβ− MCI groups than in the control group, and a further analysis of the correlations between the four test scores and cortical thickness was conducted. In the Aβ+ MCI group, the regions affected by the time effect corresponded to those affected by the main effect. For the main effect, the regions correlated with MMSE or verbal fluency scores were spread across a wider area relative to those correlated with STM or orientation (Figure 4). The hippocampal, medial prefrontal-pericallosal regions and left lateral temporal cortical degeneration were correlated with changes in STM.

In Aβ- MCI group, the regions correlated to the aforementioned cognitive test scores were topographically different from the Aβ+ MCI group. Meanwhile, the areas affected by the time effect did not correspond to the topography for the main effect. The main effect on the regions correlated with MMSE and STM were the mid-corpus callosum regions and angular gyrus. For STM, the regions affected by the main effect or time effect were in the corpus callosum. In the Aβ− MCI group, the degeneration of regions related to verbal fluency were observed in the supplementary motor areas, and degeneration of the parietal, hippocampus, and dorsolateral prefrontal areas corresponded to lower orientation scores (Figure 5).

## 3. Discussion

### 3.1. Major Findings

This study compared the differences in the surface morphometries and cognitive decline trajectories of the Aβ+ and Aβ− MCI groups and delineated their degeneration patterns on the basis of amyloid burden. The cross-sectional comparisons conducted with regard to MCI stage revealed lower STM, orientation, and verbal fluency scores in the Aβ+ and Aβ− groups relative to the control group, indicating the existence of clinical AD phenotypes in those without amyloid deposition. A higher proportion of *APOE* ε4 carrier was detected in the Aβ+ group than in Aβ− MCI group, and the effect of *APOE* ε4 on Aβ+ cortical regions acted in synergy with the pathological process. The cortical atrophy in the Aβ− MCI group was prominent in the posterior brain region, and cognitive progression was slow. *APOE* ε4 status did not have a clear effect on Aβ− cortical degeneration areas; however, the results suggest a possible link between *APOE* ε4 status and cortical vulnerability. Finally, the cortical–cognitive relationship between Aβ+ and Aβ− MCI patients suggest that different neurobiological mechanisms mediate clinical manifestations.

### 3.2. Cognitive Indicators of Amyloid Deposition as Revealed through Longitudinal Observation

The patients with Aβ+ MCI obtained an initial MMSE score of 23. A domain-specific decline was observed, suggesting that longitudinal changes in both STM and orientation subscores are sensitive clinical indicators of amyloid deposition in patients with MCI. Two studies have reported on the roles of various cognitive tests in Aβ+ MCI. Doraiswamy et al. suggested that patients with both MCI and amyloid pathology exhibit a decline in their MMSE scores and Alzheimer’s Disease Assessment Scale Cognitive subscale scores after 36 months [15]. In one study, patients with Aβ+ status exhibited a significant decline in all aspects of memory and nonmemory function over a 3-year follow-up period [16]. On the basis of on our results, the decrease in both STM and orientation scores after one year suggests that the patients with MCI were amyloid-positive.

Our Aβ− group did not exhibit significant cognitive decline; this finding is consistent with the literature on suspected non-Alzheimer’s disease pathophysiology (SNAP). SNAP is regarded as a pathological entity that comprises heterogenous neurodegenerative diseases [17]. A study on SNAP reported retrieval problems that required clinical attention [18]. If a retrieval deficit detected through an episodic memory test (e.g., CVLT) at baseline does not allow for Aβ− MCI to be differentiated from Aβ+ MCI, a longitudinal follow-up can still be performed to achieve differentiation. The Aβ+ group exhibited a decline in all cognitive tests, whereas the Aβ− group only exhibited a decline in cue-correct and orientation.

### 3.3. Recapturing Neurodegeneration through MRI in Aβ+ MCI Group

Several studies have used structural MRI to model the effect of amyloid on MCI. In patients with MCI, an amyloid load may result in initial parietotemporal cortical atrophy [19] in the precuneus, supramarginal, inferior parietal, hippocampus, and superior temporal regions and result in subsequent atrophy in the frontal lobe regions [20]. Our LME model revealed yearly cortical thinning in the Aβ+ group. The time effect (Figure 1B,C) suggests that the Aβ+ group started to undergo cortical thinning in the right temporal region, followed by thinning in the entorhinal, precuneus, medial prefrontal subgenual region, and anterior temporal areas. The hippocampal–entorhinal axis represents the regions that are affected early by amyloid deposition. These areas may exhibit epicenter properties in amyloid-β cascades [21].

Relative to the Aβ+ group, less extensive regions of cortical thinning were detected in the Aβ− group, and the cortical thinning was more prominent in the posterior cortical regions. Most of these areas were within the Aβ+ MCI degeneration regions (Figure 1B, AD time effect) [22]. However, our data suggest differences in the degenerative trajectories of Aβ− MCI relative to those of Aβ+ MCI (Figure 3A3). These differences may help explain the differences in the clinical features of the Aβ+ and Aβ− MCI groups. The pattern of atrophy over time in the Aβ− group is not solely attributable to the aging process because the Aβ− group still exhibited more atrophy over time relative to the age-matched controls (Figure 3A2).

### 3.4. Pathological Basis of Aβ− MCI and Possible Differential Diagnosis

In the current study, Aβ- group can only be regarded as a slowly progressive disorder with amnestic features mimicking Aβ+ group. Because SNAP comprises several pathological processes [22], the conditions of the Aβ− group can be considered to be from the differential list of SNAP and regarded as an amnestic form of SNAP with MCI-like clinical features. The progression rate of SNAP varies from 0% to 56% within a 3-year interval time [23,24,25]; disease progression is even faster than that of AD [23]. The Aβ− group had a stable degenerative disease from both clinical and neuroimaging perspectives. For the Aβ− group, the pathological substrates that contributed to their cognitive impairment or cortical thinning should be determined. This category comprises several different types of amyloid-unrelated pathologies such as primary age–related tauopathy (PART) [26], hippocampal sclerosis [27,28,29], TAR DNA binding protein (TDP)-43 pathology [30,31], argyrophilic grain disease (AGD) [32,33], and accelerated aging [34]. The clinical features of the Aβ− group corresponded to the finding of another study that a nonnegligible proportion of patients clinically diagnosed as having MCI due to AD did not have amyloid deposition in their histopathology [4].

Medial temporal atrophy was reported in patients with PART [26], hippocampal sclerosis [27,28,29], and AGD [32,33]. In our study, the patients with Aβ− status did not exhibit any atrophy in the medial temporal lobe, suggesting a low proportion of patients within this category. In addition, medial frontal cortical atrophy is associated with the topographic features of atrophy in frontotemporal lobar degeneration (FTLD)-TDP [31]. In our study, the patients with Aβ− status did not exhibit significant medial frontal cortical thinning; thus, they did not fit into this category. Frontotemporal atrophy occurred with accelerated aging in the regions that involved the medial and lateral temporal lobes, the medial and lateral frontal cortices, and the precuneus/retrosplenial cortex [34]. Some of the patients with Aβ− status who exhibited frontal and temporal atrophy could have an accelerated aging pathogenesis.

The lack of a diagnostic clinical phenotype and clinical diagnostic criteria for patients of Aβ− status limited our ability to obtain findings that can contribute to antemortem diagnosis and the development of specific treatment strategies. The major focus on reporting the Aβ- MCI cross- or longitudinal trajectory was based on search strategies for patients requiring amyloid removal therapy. A study reported four trajectories of tau deposition in patients with AD; specifically, 30.5% of its AD population exhibited an epicenter in the posterior brain region [35]. A further evaluation is required to determine whether the patients with Aβ− status experienced very-early-stage AD, resulting in a negative amyloid status. Molecular neuroimaging should clarify the status of the pathological substrates in our Aβ− group.

### 3.5. Effect of *APOE* ε4 on Aβ+ or Aβ− MCI at Baseline or Longitudinal Cortical Degeneration

Our study revealed that 52% of the patients with Aβ+ status were *APOE* ε4 carriers. The main effect pertained to cortical vulnerability, whereas the time effect pertained to *APOE* ε4–related cortical degeneration. In the Aβ+ group, *APOE* ε4 cortical vulnerability was spread across the cortex and had low diagnostic value; by contrast, the effect of *APOE* ε4 on the degenerative process was widely dispersed (Figure 3B,C in Aβ+ group) and stronger than the Aβ+ degenerative pattern. Therefore, the *APOE* ε4 allele might have synergized with other genes [36] or pathological cascades [37,38] in the Aβ+ group. This finding is consistent with the those of other studies, which have considered Aβ status in predictions of disease trajectories and have reported that *APOE* ε4 generally augments the pathological cascades of neurodegeneration [15,39,40].

Despite the heterogeneous etiologies of the patients with Aβ− status, the detrimental effects of *APOE* were still considered. In our Aβ− cohort, the proportion of *APOE* ε4 carriers was 13%, and the time effect of *APOE* ε4 affected regions that corresponded to the degenerative pattern of the Aβ− group (Figure 1B). Relative to noncarriers, the *APOE* ε4 carriers exhibited a lower cortical thickness in the DMN regions. These results suggest that *APOE* ε4 was involved in the pathogenetic mechanism of the Aβ− group.

### 3.6. Cognitive–Cortical Thickness Relationship Indicates Different Cognitive Processes of Aβ+ and Aβ− Groups

The cognitive–cortical thickness maps of the patients with Aβ+ were mainly localized in the left perisylvian areas and medial prefrontal–pericallosal regions. A study that conducted structural or functional imaging demonstrated the key roles of the left temporal and parietal perisylvian areas in patients with logopenic progressive aphasia, an atypical subtype in AD pathology [41]. However, the left perisylvian region is involved in STM, verbal fluency, and orientation, and it influences MMSE results. Koenigs et al. reported on the key neural substrates underlying verbal STM and language processing abilities [42]; they also revealed that the left perisylvian cortex is related to auditory–verbal STM performance. Peer et al. reported that the regions over the inferior parietal lobe and temporal lobe near the perisylvian region are active during space-, time-, and person-related orientation [43].

The cognitive–cortical map of the Aβ− group was scattered in the supramarginal, lateral temporal, genu of the anterior cingulate, and hippocampus. The different topographic correlations between cortical atrophy and cognitive decline in the Aβ+ and Aβ− groups indicate the presence of different mechanisms in the two disease spectrums; the lack of correlations in the Aβ− group indicates a stable condition.

### 3.7. Limitations

This study has several limitations. First, all our patients with and without amyloid deposition met the core clinical criteria for MCI due to AD at the time of enrollment. Both the Aβ+ and Aβ− MCI groups had single-amnestic or multidomain-amnestic subtype [4,39], and they could have exhibited different progression rates. Second, the sample size used in our analysis was small; our results should be generalized to other Mandarin-speaking cohorts. Nonetheless, this study implemented a long follow-up period, and our results indicate that only longitudinal data can help differentiate amyloid-positive status from amyloid-negative status. Third, although amyloid pathology was likely developed at the preclinical and prodromal stages over numerous years, we focused on the patients with MCI due to AD and only monitored the level of amyloid at baseline. Consequently, we could have excluded patients who experienced amyloid development during the follow-up period. Therefore, future studies should reexamine amyloid deposition during follow-up. Fourth, amyloid-negative MCI is a clinical definition and pathologically heterogeneous. Future studies can use molecular imaging to further clarify the proportion and effect of Tau, TDP-43, or α-synuclein in patients with Aβ−. For example, our interpretations regarding clinical and structural changes should be substantiated with tau positron-emission tomography (PET) data. In one study, 17% of patients who were amyloid negative and Tau-positive exhibited *APOE* ε4 status and the features characteristic of AD [44]. Finally, because of the observational nature of the present study, we could not rule out the possible presence of unmeasured or insufficiently measured confounders.

## 4. Materials and Methods

In all, 185 participants were enrolled from the Cognitive and Aging Center of Kaohsiung Chang Gung Memorial Hospital. The study was approved by the Chang Gung Memorial Hospital Institutional Review Board, and written informed consent was obtained from all participants.

### 4.1. Group Stratification Criteria

We selected controls and patients with amnestic MCI [2] from a Cognitive and Aging (CAC) database. At baseline, all the patients underwent a demographic survey, cognitive testing, apolipoprotein E protein (*APOE*) genetic and basic blood testing, three-dimensional (3D)-T1 weighted imaging, and an amyloid scan. All the enrolled patients with amnestic MCI met the clinical diagnostic criteria for MCI due to AD [2], and their episodic memory deficits were identified on the basis of the cutoff values of the CVLT [45].

The exclusion criteria for the present study were a history of clinical stroke (*n* = 3), a modified Hachinski ischemic score of >4 (*n* = 1), degenerative brain diseases other than AD (*n* = 1), the presence of lesions on T2-weighted MRI indicating severe white matter diseases (*n* = 2), clinically unmanaged diabetes (*n* = 1), major depressive disorder (*n* = 1), and dysthymic disorder *(n* = 3) as diagnosed in accordance with the diagnostic criteria of the Diagnostic and Statistical Manual of Mental Disorders, fourth edition, text revision (DSM-IV-TR) [46]. In total, 12 patients were excluded.

The enrolled patients were subsequently stratified into the Aβ+ group or the Aβ− group on the basis of visual score readouts of two independent raters and amyloid Centiloid level of 35 or more. Accordingly, the Aβ+ MCI and Aβ− MCI groups comprised 67 and 54 patients, respectively. Healthy age- and sex-matched controls (*n* = 64) were recruited from a community source. During follow-up, the participants underwent annual cognitive tests and follow-up MRI with intervals of 18–24 months from baseline.

### 4.2. Demographic Registration and Cognitive Assessment

After the patients were enrolled, their demographic data (i.e., estimated disease onset based on caregiver reports of first symptoms, years of education, gender, *APOE* ε4 status, medication, and family history) were collected. A trained neuropsychologist administered the neurobehavioral tests, which involved the use of the MMSE and CASI. The total scores for the MMSE and CASI reflect a global assessment of cognitive function. The CASI contains nine subdomains that are used to assess various executive functions, including attention, verbal fluency, abstract thinking, and mental manipulation [47]. In the present study, orientation, short- and long-term memory, language ability, and drawing were regarded as nonexecutive domains. Because the patients with Aβ+ and Aβ− status had salient memory complaints, we also used the CVLT to assess their verbal episodic memory [45]. The CVLT comprises four learning trails, which are followed by a 30-s recall test (30-s recall), a 10-min recall test (10-min recall), a cued recognition test, and a cue recall test.

### 4.3. APOE Genotyping

Single nucleotide polymorphism genotyping was performed using a MassARRAY system with iPLEX Gold chemistry (Agena Bioscience, San Diego, CA, USA). Extended polymerase chain reaction products were purified using cation exchange resins and then spotted onto a 384-format SpectroCHIP II array by using a MassArray Nanodispenser RS1000. Mass determination was performed using a MassARRAY compact analyzer. The resulting spectra were processed, and alleles were called by using a MassARRAY Typer 4.0 and performing a model-based cluster analysis to analyze the genotypes of single nucleotide polymorphisms. The *APOE* genotype was determined using rs7412 and rs429358. A *APOE* ε4 carrier was defined as a participant with one or two ε4 alleles. The obtained genotypes were dichotomized into ε4 carriers (heterozygous or homozygous) and noncarriers (i.e., ε2 or ε3 carrier).

### 4.4. MR image Acquisition and Processing

MR images were obtained using a 3T GE Discovery 750 device (GE Medical Systems, Milwaukee, WI, USA). All MRI scans were processed using a workstation (Macintosh iMac Pro 2017, MacOS Catalina, version 10.15.16) and the FreeSurfer image analysis suite v7.1.1 (http://surfer.nmr.mgh.harvard.edu (accessed on 4 September 2022)) [48]. Details on MRI data acquisition and processing procedures are described in Appendix A.

### 4.5. Amyloid Image Acquisition and Processing

The amyloid tracer [^18^F]flubetaben or [^18^F]florbetapir was used. Brain PET scans were acquired after the injection of 296 ±74 MBq by using a GE discovery MI PET/CT scanner. PET image acquisition consisted of two 5-min dynamic frames to allow for motion correction; 3D PET images were acquired and images were reconstructed with an interative reconstruction algorithm (OSEM 4 iterations, 16 subsets) and a post hoc 5 mm Gaussian filter. Low-dose CT scans for attenuation correction with the following parameters were acquired: 15 mAs, 120 keV, 512 × 512 matrix, 2.79-mm slice thickness, 71 slices, 110-mm/s increment, 0.5-s rotation time, and pitch of 1.375. One 3D T1 images corresponding to the time point of amyloid PET was used for partial volume correction and for the Centiloid calculation. The quantification steps for Centiloid scale followed those described by the Centiloid project (www.gaain.org (accessed on 4 September 2022)).

### 4.6. Statistical Analyses

Cross-sectional cognitive data for differentiating between the three groups were analyzed by performing analyses of variance; subsequently, Bonferroni corrections and chi-square tests were performed for categorical data. For the time effect, we calculated the intervals (months) between disease onset and cognitive test date. For the control group, disease duration was defined as the number of months following the baseline visit. For longitudinal cognitive data, we applied LME models and used group, time, and group–time interaction to assess cognitive changes. Details on the trajectory of cognitive function and MRI analysis are presented in Appendix A.

## 5. Conclusions

In this study, Aβ+ MCI was differentiated from Aβ− MCI on the basis of longitudinal cognitive measurements and the cortical thickness model. Compared with the Aβ− and control groups, the Aβ+ group exhibited prominent and greater rates of cognitive decline and cortical thinning. Furthermore, *APOE* ε4 was revealed to be capable of augmenting cortical atrophy in patients with Aβ+ MCI. Differentiating between these two disease spectrums on the basis of cross-sectional measurements is challenging; however, data pertaining to *APOE* ε4 status, cortical thinning trajectories, and cognitive follow-up can be combined and used to differentiate between these two diseases.

## Figures and Tables

**Figure 1 ijms-23-14635-f001:**
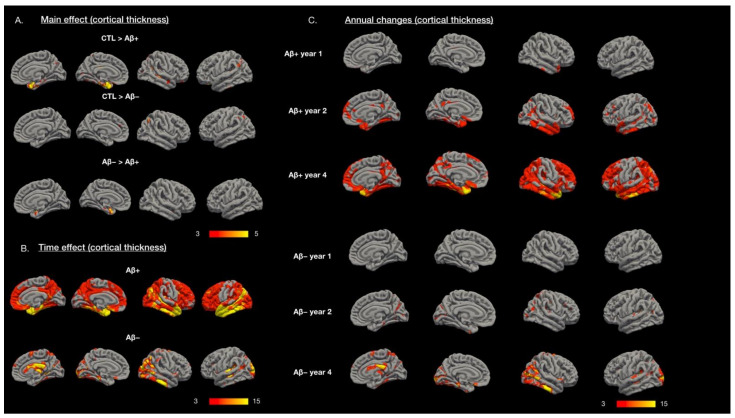
(**A**) Significant group effects in cortical thickness comparing amyloid-positive (Aβ+) and amyloid-negative (Aβ−) mild cognitive impairment and age-matched controls (CTL). (**B**) Cortical thickness degenerative topography in Aβ+ and Aβ− showed distinct patterns using longitudinal mixed effect model. (**C**) Annual changes in cortical thickness in Aβ+ and Aβ− groups. Color bar numbers = −log_10_P, P = *p* value.

**Figure 2 ijms-23-14635-f002:**
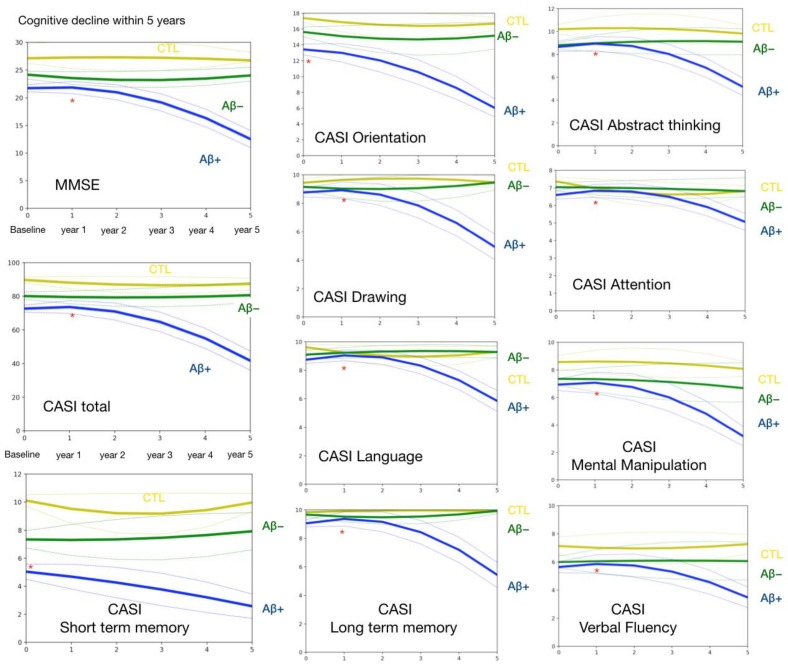
Longitudinal cognitive trajectory using mini-mental state examination (MMSE) and cognitive ability screening instrument (CASI) total or subdomains. The solid trendline uses a 2nd-degree polynomial fit, the dashed line represents 95% confidence intervals, asterisks represent the turning point of the trendline in Aβ+. Mild cognitive impairment (MCI) with amyloid-positive (Aβ+), MCI with amyloid-negative (Aβ−), age-matched controls (CTL).

**Figure 3 ijms-23-14635-f003:**
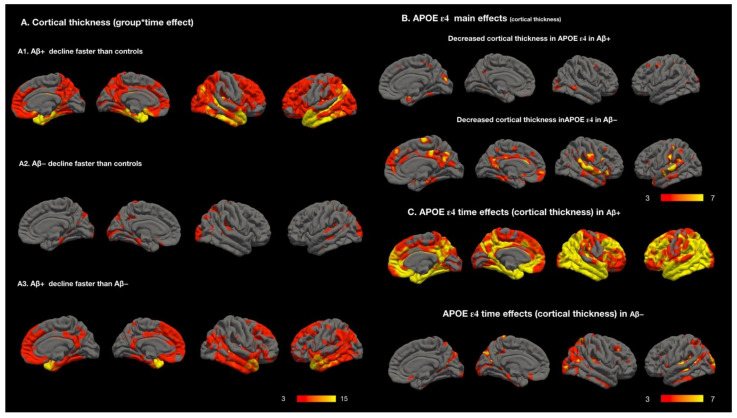
(**A**) Group–time interactions in cortical thickness comparing degenerative rate of amyloid-positive group (Aβ+), amyloid-negative group (Aβ−), and controls. (**B**) Decreased cortical thickness in *APOE* ε4 carrier compared with noncarrier main effects. (**C**) Decreased cortical-thickness–time interaction in *APOE* ε4 carrier compared with noncarrier in Aβ+ and Aβ−. Color bar numbers = −log_10_P, P = *p*-value.

**Figure 4 ijms-23-14635-f004:**
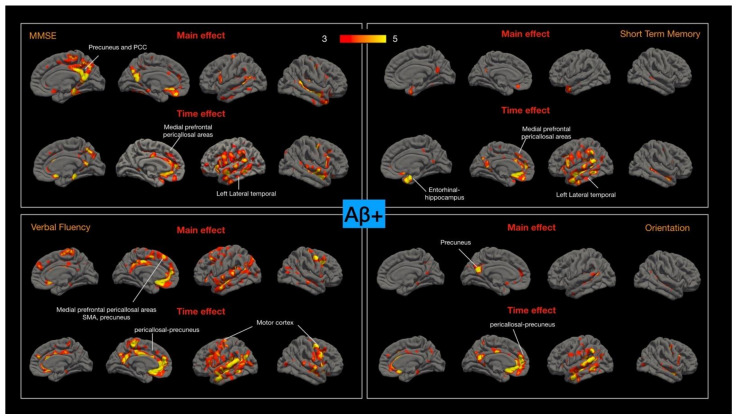
Cortical-thickness–cognitive main effects or time effects in amyloid-positive group (Aβ+). Four cognitive tests were selected based on the baseline cognitive test scores significance with controls. Color bar numbers = −log_10_P, P = *p* value.

**Figure 5 ijms-23-14635-f005:**
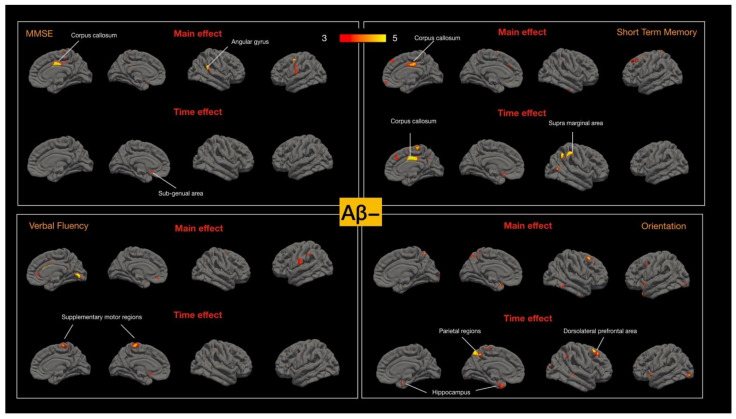
Cortical-thickness–cognitive main effects or time effects in amyloid-negative group (Aβ−). Four cognitive tests were selected based on the baseline differences with controls. Color bar numbers = −log_10_P, P = *p* value.

**Table 1 ijms-23-14635-t001:** Participant characteristics by clinical groups.

	Cognitive Unimpaired Controls *n* = 64	Mild Cognitive Impairment Due to Alzheimer’s Disease *n* = 121	
Name in the manuscript	Controls	Aβ− MCI	Aβ+ MCI
Amyloid classification	Amyloid Negative	Amyloid Negative, *n* = 54	Amyloid Positive, *n* = 67
Education (years)	8.5 (5.4)	9.3 (3.8)	8.3 (4.5)
Onset Age (year-old)	N.A	67.1 (7.4)	67.2 (7.3)
Age at enrollment (year-old)	68 (6.5)	69.2 (7.6)	68.9 (7.1)
Gender (female/male)	40/24	30/24	40/27
Hypertension, *n* (%)	30 (46.9)	22 (40.7)	29 (43.3)
Diabetes Mellitus, *n* (%)	28 (43.8)	18 (33.3)	14 (20.9)
Hyperlipidemia, *n* (%)	20 (31.3)	12 (22.2)	20 (29.9)
Apolipoprotein E4, *n* (%)	10 (15.6)	7 (13)	35 (52)
Baseline mini-mental state examination	26.8 (3.1)	24.5 (2.9) *	22.9 (4.2) *^,+^
Baseline CASI_Total	88.5 (9.2)	80.6 (9.8) *	75.3 (14.6) *
Mental manipulation (10)	8.1 (2.4)	7.4 (2.5)	7.4 (2.7)
Attention (8)	7.2 (0.9)	7.1 (1.0)	6.8 (1.2)
Orientation (18)	17.5 (1.3)	15.8 (2.7) *	14.2 (4.3) *
Long-term memory (10)	9.7 (1.3)	9.7 (0.7)	9.3 (1.6)
Short-term memory (12)	10.2 (2.2)	7.4 (3.0) *	5.3 (3.3) *^,+^
Abstract thinking (12)	10 (1.8)	8.9 (2)	8.9 (2.3)
Drawing (10)	9.4 (1.2)	9.1 (1.5)	9.1 (1.8)
Verbal fluency (10)	7.2 (2.3)	5.9 (2.3) *	5.8 (2.6) *
Language (10)	9.5 (1.2)	9.3 (1.2)	8.9 (1.6)
CVLT total scores			
CVLT_T1 (9)	4.07 (1.77)	3.64 (1.11)	3.27 (1.39) *
CVLT_T2 (9)	5.59 (1.55)	5.11 (1.26)	4.80 (1.70)
CVLT_T3 (9)	6.41 (1.47)	5.77 (1.45)	5.36 (1.77) *
CVLT_T4 (9)	7.04 (1.26)	6.17 (1.39)	5.66 (1.74) *
30 s (9)	6.74 (1.68)	5.11 (2.12) *	4.15 (2.33) *
10 min (9)	6.04 (2.03)	3.85 (2.76) *	2.63 (2.80) *
Cue recall (9)	6.22 (2.12)	4.15 (2.66) *	2.61 (2.82) *
Cue correct (9)	8.11 (1.48)	7.64 (1.88)	6.78 (2.76) *

Abnormal amyloid PET was defined by positive reading by two nuclear medicine experts and Centiloid level of 35 or more. Data represent mean (standard deviation). * *p* < 0.05, compared with cognitively unimpaired controls; ^+^ *p* < 0.05, between two MCI groups. CVLT: Chinese version Verbal Learning Test; CASI: cognitive ability screening instrument; MCI: mild cognitive impairment; Aβ+ MCI: MCI due to Alzheimer disease (AD) with positive amyloid status; Aβ− MCI: MCI with negative amyloid status.

**Table 2 ijms-23-14635-t002:** Summary of cognitive decline in differentiating two MCI groups referenced to controls.

		Main Effect			Group–Time Interaction	
		Aβ+	Aβ−	Others	Aβ+	Aβ−
General	Mini-mental state examination	+	+		+	−
	CASI	+	+	Edu	+	−
Memory	Short-term memory	+	+		+	−
	California Verbal Learning Test, 10-min recall	+	−		+	−
	California Verbal Learning Test, cue-correct	+	−		+	+
CASI Executive	Verbal fluency	−	−	Edu	+	−
	Abstract thinking	−	+	Edu	+	−
	Mental manipulation	−	−	Edu	+	−
	Attention	−	−	Edu	+	−
CASI Non-Executive	Orientation	+	+	Edu × Aβ−	+	+
	Language	−	−	Edu	+	−
	Drawing	−	−	Edu	+	−
	Long-term memory	+	+		+	−

+ *p* < 0.05 represents statistical significance in the main effect (controls as reference) or significance in the time effect, while − represents no statistical significance. Aβ+: amyloid-positivity; Aβ−: amyloid-negative group. Edu: educational year. MCI: mild cognitive impairment, CASI: cognitive ability screening instrument.

## Data Availability

The data that support the findings of this study are available on request from the corresponding author. The data are not publicly available due to privacy or ethical restrictions.

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
