# Peer review of "Impact of Amyloid Pathology in Mild Cognitive Impairment Subjects: The Longitudinal Cognition and Surface Morphometry Data"

_ijms, 2022, doi:10.3390/ijms232314635_

Round 1
Reviewer 1 Report
This is one of the best papers I read in 2022. The experimental design is straightforward and the data analysis is excellent. I only have one suggestion, which might further improve this outstanding manuscript: The authors should analyze the data by gender. In other words, are the Abeta+ males different from the Abeta+ females? The same question applies to the Abeta- group.
Author Response
Response to Reviewer 1 Comments
Point 1: This is one of the best papers I read in 2022. The experimental design is straightforward and the data analysis is excellent. I only have one suggestion, which might further improve this outstanding manuscript: The authors should analyze the data by gender. In other words, are the Abeta+ males different from the Abeta+ females? The same question applies to the Abeta- group.
Response 1:
We appreciate the reviewer providing such suggestions. We accessed the gender effects from the baseline cognitive test using CASI total or CASI short term memory (STM) as outcome variables. For the Ab+ group, there were significantly higher education levels in the male individuals, so the educational year was used for adjusted covariates (female mean 6.9-year, male mean 9.42year, p=0.002). In the Ab- group, there were no educational differences (female mean 8.59, male 9.97 year).
We first explore the baseline cognitive tests score by gender. The gender effect was not significant in CASI total (b=-3.972, p=0.47) or CASI STM (b=-2.564, p=0.47) scores in the Ab+ group. The educational levels were related to higher CASI scores (b=0.912, p=0.003). In Ab- group, there was also no gender effect in CASI total (b=0.037, p=0.987) or CASI STM (b=-0.225, p=0.739) scores.
For the longitudinal data in the Ab- group, there was no gender effect (CASI b=-3.755, p=0.279, STM b=-0.810, p=0.313, male as the reference). For the longitudinal data in the Ab+ group, there was no gender effect (CASI total b=-6.064, p=0.138, STM b=-0.323, p=0.75, male as the reference).
Based on the findings, we considered no gender effect either in Ab+ or Ab- group on cross sectional or longitudinal salient cognitive features. The results were included in ‘Result section 2.1’ and ‘Result section 2.3.’ as follows:
“We assessed the effect of gender on the cognitive measurements of the two MCI groups and discovered that gender did not have an effect on such measurements in the Ab+ group (total CASI [b = −3.972, p = 0.47]; STM [b = −2.564, p = 0.47)) and Ab− group (total CASI [b = 0.037, p = 0.987]; STM (b = −0.225, p = 0.739)).” (page 3, line 111); and “We assessed the effect of gender on longitudinal cognitive decline in the two MCI groups and discovered that gender did not influence the results of the Ab+ group (total CASI [b = −6.064, p = 0.138, STM b = −0.323, p = 0.75]; male gender as a reference) and Ab− group (CASI [b = −3.755, p = 0.279, STM b = −0.810, p = 0.313]; male gender as a reference). ” (page 5, line 150)

Reviewer 2 Report
The authors performed an interesting study that includes a longitudinal analysis in a Mandarin-speaking cohort of patients with amnestic mild cognitive impairment (MCI) at multiple levels (clinical, cognitive, MRI, and amyloid PET) with relevant results. The study is interesting and deserves publication but needs improvement in several aspects. First, it requires improvement in English language since it is very hard to read as it is.
Another suggestion is that, in my view, the classification of the patients with cognitive impairment included in the study should be “amnestic MCI” instead of MCI of the AD type, since this last term is reserved for patients with positive biomarkers, and therefore in this case it would apply only to the Ab positive group.
The abstract should indicate the tracer used for amyloid PET.
The introduction should include a paragraph clearly stating what is new in this study.
In Results Table 1 needs improvement to make it easier to understand:
- The units of measurement should be included in parenthesis in the first column
- Categorical variables, would be better indicated as number and percent (n,%), instead of no/yes. (In “gender” in doesn’t indicate which is female or male)
- In Table 1 and the other tables the p value and statistical method used should be stated.
Abbreviations should be defined the first time they appear (they are defined in Methods, but this is the last section). eTIV is not defined.
In Methods the exclusion criteria are defined twice (in section 4.1 Group stratification criteria and in section 4.2. Demographic registration and cognitive assessment) and they do not match. Methods should indicate what PET instrument and software were used. The number of missing and drop out data are not revealed. The method to diagnose depression or determine ApoE status are not stated.
The conclusion paragraph starts with a sentence that is not the main result. I suggest they rewrite it emphasizing the main results found in Ab+ and Ab- groups, in addition to what it is said.
Minor corrections:
line 106: It should say "compared with amnestic group with negative amyloid scan"
Paragraph 2.6, lines 174-185 are confusing, please write more clearly.
Lines 190-191: It seems to me that in the sentence: “were topographically wider in which the areas related to short term memory and orientation (Figure 4)”, should be “were topographically wider than in areas related to short term memory and orientation (Figure 4)”
Author Response
Response to Reviewer 2 Comments
Major corrections:
Point 1: First, it requires improvement in English language since it is very hard to read as it is.
Response 1:
Thanks for the kind reminder. We have sent the English language edit to improve reading and rewrote several sentences in the whole manuscript. We also added a sentence in the ‘Acknowledgments’ section as follows: “This manuscript was edited by Wallace Academic Editing.” (page 15, line 512).
Point 2: Another suggestion is that, in my view, the classification of the patients with cognitive impairment included in the study should be “amnestic MCI” instead of MCI of the AD type, since this last term is reserved for patients with positive biomarkers, and therefore in this case it would apply only to the Ab positive group.
Response 2:
Thanks for the kind suggestion. We enrolled MCI patients based on the “clinical criteria” of ‘MCI due to AD’ ; then we divided MCI patients into Aβ- or Aβ+MCI group by amyloid PET. We rewrote two sentences in the first paragraph of “4.1. Group stratification criteria” section as follows:
“We selected controls and patients with amnestic MCI [2] from a Cognitive and Aging (CAC) database.” (page 13, line 398) and “All the enrolled patients with amnestic MCI met the clinical diagnostic criteria for MCI due to AD [2], and their episodic memory deficits were identified on the basis of the cutoff values of the CVLT [45].” (page 13, line 401)
Point 3: The abstract should indicate the tracer used for amyloid PET.
Response 3:
We rewrote a sentence in the ‘Abstract’ section as follows:
“The patients with MCI were classified into two groups on the basis of their [18F]flubetaben or [18F]florbetapir amyloid positron-emission-tomography scan (Aβ+ vs. Aβ−, 67 vs. 54 patients) results.” (page 1, line 27)
Point 4: The introduction should include a paragraph clearly stating what is new in this study.
Response 4:
We acknowledge the vital suggestion. We add three sentences in the last paragraph of ‘Introduction’ as follows:
“We hypothesized that longitudinal cognition and surface morphometric changes could help differentiate between Aβ+ and Aβ− MCI groups. Direct comparisons of these two groups can help delineate degeneration patterns by amyloid burden. The differentiation of these two groups is crucial to the development and implementation of treatment strategies.” (page 2, line 78) .
Point 5: In Results Table 1 needs improvement to make it easier to understand:
- The units of measurement should be included in parenthesis in the first column
- Categorical variables, would be better indicated as number and percent (n,%), instead of no/yes. (In “gender” in doesn’t indicate which is female or male)
- In Table 1 and the other tables the p value and statistical method used should be stated.
Response 5:
Thanks for your kind reminder. We modified tale 1 by your suggestion as follows:
Table 1 Participant characteristics by clinical groups
|
|
Cognitive Unimpaired Controls n=64 |
Mild Cognitive Impairment due to Alzheimer’s Disease n=121 |
|
|
Name in the manuscript |
Controls |
Aβ- MCI |
Aβ+ MCI |
|
Amyloid classification |
Amyloid Negative |
Amyloid Negative, n=54 |
Amyloid Positive, n=67 |
|
Education (years) |
8.5 (5.4) |
9.3 (3.8) |
8.3 (4.5) |
|
Onset Age (year-old) |
N.A |
67.1 (7.4) |
67.2 (7.3) |
|
Age at enrollment (year-old) |
68 (6.5) |
69.2 (7.6) |
68.9 (7.1) |
|
Gender (female/male) |
40/24 |
30/24 |
40/27 |
|
Hypertension, n (%) |
30 (46.9) |
22 (40.7) |
29 (43.3) |
|
Diabetes Mellitus, n (%) |
28 (43.8) |
18 (33.3) |
14 (20.9) |
|
Hyperlipidemia, n (%) |
20 (31.3) |
12 (22.2) |
20 (29.9) |
|
Apolipoprotein E4, n (%) |
10 (15.6) |
7 (13) |
35 (52) |
|
Baseline Mini-mental state examination |
26.8 (3.1) |
24.5 (2.9)* |
22.9 (4.2)*+ |
|
Baseline CASI_Total |
88.5 (9.2) |
80.6 (9.8)* |
75.3 (14.6)* |
|
Mental manipulation (10) |
8.1 (2.4) |
7.4 (2.5) |
7.4 (2.7) |
|
Attention (8) |
7.2 (0.9) |
7.1 (1.0) |
6.8 (1.2) |
|
Orientation (18) |
17.5 (1.3) |
15.8 (2.7)* |
14.2 (4.3)* |
|
Long term memory (10) |
9.7 (1.3) |
9.7 (0.7) |
9.3 (1.6) |
|
Short term memory (12) |
10.2 (2.2) |
7.4 (3.0)* |
5.3 (3.3)*+ |
|
Abstract thinking (12) |
10 (1.8) |
8.9 (2) |
8.9 (2.3) |
|
Drawing (10) |
9.4 (1.2) |
9.1 (1.5) |
9.1 (1.8) |
|
Verbal fluency (10) |
7.2(2.3) |
5.9 (2.3)* |
5.8 (2.6)* |
|
Language (10) |
9.5(1.2) |
9.3 (1.2) |
8.9 (1.6) |
|
CVLT total scores |
|||
|
CVLT_T1 (9) |
4.07(1.77) |
3.64(1.11) |
3.27(1.39)* |
|
CVLT_T2 (9) |
5.59(1.55) |
5.11(1.26) |
4.80(1.70) |
|
CVLT_T3 (9) |
6.41(1.47) |
5.77(1.45) |
5.36(1.77)* |
|
CVLT_T4 (9) |
7.04(1.26) |
6.17(1.39) |
5.66(1.74)* |
|
30 sec (9) |
6.74(1.68) |
5.11(2.12)* |
4.15(2.33)* |
|
10min (9) |
6.04(2.03) |
3.85(2.76)* |
2.63(2.80)* |
|
Cue recall (9) |
6.22(2.12) |
4.15(2.66)* |
2.61 (2.82)* |
|
Cue correct (9) |
8.11(1.48) |
7.64(1.88) |
6.78(2.76)* |
(page 3, line 116).
We stated the p value and statistical method in the footnote of ‘Table 1’ and other Tables as follows:
Table 1: * p<0.05, compared with cognitively unimpaired controls; + p<0.05, between two MCI groups. (page 3, line 119)
Table 2: + p<0.05, represents statistical significance in main effect (controls as reference) or significance in time effect, while – represents no statistical significance. (page 6, line 165)
Point 6: Abbreviations should be defined the first time they appear (they are defined in Methods, but this is the last section). eTIV is not defined.
Response 6:
Thanks for the kind reminder. We defined the abbreviations MMSE, CASI, and CVLT in the second paragraph of ‘2.1. Baseline demographics and cognitive tests’ section as follows:
“At baseline, the mini-mental state examination (MMSE) and cognitive abilities screening instrument (CASI) scores of the Aβ+ and Aβ− MCI groups were significantly lower than those of the controls (Table 1)” (page 3, line 100); and “With respect to Chinese-version verbal learning test (CVLT) results at baseline, the Aβ+ group obtained significantly lower scores in the learning trials (T1–T4), 30-s and 10-min recall tests, and cue-correct and cue-recall test.” (page 3, line 105)
We rewrote two sentences in the “4.2. Demographic registration and cognitive assessment” as follows:
“A trained neuropsychologist administered the neurobehavioral tests, which involved the use of the MMSE and CASI” (page 14, line 420 ); and “Because the patients with Aβ+ and Aβ− status had salient memory complaints, we also used the CVLT to assess their verbal episodic memory [45].” (page 14, line 426).
We already defined the eTIV as “estimated total intracranial volume” in the first paragraphy of ‘2.2. Baseline group cortical thickness differences’ section. (page 4, line 126)
Point 7: In Methods the exclusion criteria are defined twice (in section 4.1 Group stratification criteria and in section 4.2. Demographic registration and cognitive assessment) and they do not match.
Response 7:
We acknowledge the vital comments. We rewrote the exclusion criteria in the second paragreaphy of ‘4.1. Group stratification criteria’ section as follows: “The exclusion criteria for the present study were a history of clinical stroke (n = 3), a modified Hachinski ischemic score of >4 (n = 1), degenerative brain diseases other than AD (n = 1), the presence of lesions on T2-weighted MRI indicating severe white matter diseases (n = 2), clinically unmanaged diabetes (n = 1), and major depressive disorder (n = 1), and dysthymic disorder (n = 3) as diagnosed in accordance with the diagnostic criteria of the Diagnostic and Statistical Manual of Mental Disorders, fourth edition, text revision (DSM-IV-TR) [46]” (page 13, line 404)
Point 8: Methods should indicate what PET instrument and software were used.
Response 8:
Thanks for the kind reminder. The PET acquisition and quantification for Centiloid scales is included in section 4.5. as follows:
“4.5. Amyloid image acquisition and processing
The amyloid tracer [18F]flubetaben or [18F]florbetapir was used. Brain PET scans were acquired after the injection of 296 ±74 MBq by using a GE discovery MI PET/CT scanner. PET image acquisition consisted of two 5-minute dynamic frames to allow for motion correction; 3D PET images were acquired and images were reconstructed with an interative reconstruction algorithm (OSEM 4 iterations, 16 subsets) and a post hoc 5 mm Gaussian filter.. Low-dose CT scans for attenuation correction with the following parameters were acquired: 15 mAs, 120 keV, 512 × 512 matrix, 2.79-mm slice thickness, 71 slices, 110-mm/s increment, 0.5-s rotation time, and pitch of 1.375. One 3D T1 images corresponding to the time point of amyloid PET was used for partial volume correction and for the Centiloid calculation. The quantification steps for Centiloid scale followed that described in the Centiloid project (www.gaain.org).” (page 14, line 449).
Point 9:The number of missing and drop out data are not revealed.
Response 9:
Thanks for the kind reminder. As these cases were enrolled from the cognitive and aging database, we extracted all the available data for analysis using linear mixed effect model. During the screening phase and the exclusion criteria, we have excluded 12 patients. In result section 2.1, we reported that the patients had a mean follow-up month of 40 months (minimal 13, maximal 187). Based on the follow-up, all patients had at least two cognitive tests. To consider for the repeated measurements and the dependency of data within a subject, we used linear mixed effect model (LME). The advantage of using LME is that the structural model is based on the population, and not on data from any particular subject, thus allowing for sparse sampling. In addition, the between- and within-subject variability may be estimated. Therefore, we did not report the drop out data (or missing data) in the material and method section.
Point 10:The method to diagnose depression or determine ApoE status are not stated.
Response 10:
Thanks for the kind reminder. We rewrote the exclusion criteria in the second paragreaphy of ‘4.1. Group stratification criteria’ section to diagnose depression as follows:
“The exclusion criteria for …… and major depressive disorder (n = 1), and dysthymic disorder (n = 3) as diagnosed in accordance with the diagnostic criteria of the Diagnostic and Statistical Manual of Mental Disorders, fourth edition, text revision (DSM-IV-TR) [46].” (page 13, line 409)
We added a section in the ‘4.3. APOE genotyping’ as follows:
“4.3. APOE genotyping
Single nucleotide polymorphism genotyping was performed using a MassARRAY system with iPLEX Gold chemistry (Agena Bioscience, San Diego, CA, USA). Extended polymerase chain reaction products were purified using cation exchange resins and then spotted onto a 384-format SpectroCHIP II array by using a MassArray Nanodispenser RS1000. Mass determination was performed using a MassARRAY compact analyzer. The resulting spectra were processed, and alleles were called by using a MassARRAY Typer 4.0 and performing a model-based cluster analysis to analyze the genotypes of single nucleotide polymorphism. The APOE genotype was determined using rs7412 and rs429358. A APOE ε4 carrier was defined as a participant with one or two ε4 alleles. The obtained genotypes were dichotomized into ε4 carriers (heterozygous or homozygous) and noncarriers (i.e., ε2 or ε3 carrier).” (page 14, line 431)
Point 11: The conclusion paragraph starts with a sentence that is not the main result. I suggest they rewrite it emphasizing the main results found in Ab+ and Ab- groups, in addition to what it is said.
Response 11:
Thanks for the kind suggestion. We rewrote the ‘5. Conclusions’ sentence as follows:
“In this study, Aβ+ MCI was differentiated from Aβ− MCI on the basis of longitudinal cognitive measurements and the cortical thickness model. Compared with the Aβ− and control groups, the Aβ+ group exhibited prominent and greater rates of cognitive decline and cortical thinning. Furthermore, APOE ε4 was revealed to be capable of augmenting cortical atrophy in patients with Aβ+ MCI. Differentiating between these two disease spectrums on the basis of cross-sectional measurements is challenging; however, data pertaining to APOE ε4 status, cortical thinning trajectories, and cognitive follow-up can be combined and used to differentiate between these two diseases.” (page 15, line 471 ).
Minor corrections:
Point 12:line 106: It should say "compared with amnestic group with negative amyloid scan"
Response 12:
Thanks for the kind reminder. We wanted to clarify that you mentioned the footnote of ‘Table 1’: ‘Aβ- MCI: MCI with negative amyloid status’ should be ‘amnestic group with negative amyloid scan’?
We answered this question in Q2 as follows: “We selected controls and patients with amnestic MCI [2] from a Cognitive and Aging (CAC) database.” (page 14, line 398) and “All the enrolled patients with amnestic MCI met the clinical diagnostic criteria for MCI due to AD [2],” (page 13, line 401)
Point 13:Paragraph 2.6, lines 174-185 are confusing, please write more clearly.
Response 13:
Thanks for the kind reminder. We rewrote the paragraph of ‘2.6. section’ as follows:
“2.6. Main and time effects of APOE ε4 on cortical thickness
In the Aβ+ MCI group, APOE ε4 carriers exhibited more scattered cortical atrophy relative to noncarriers (Figure 3B, decreased cortical thickness in APOE ε4 carriers with Aβ+ status). However, the effect of APOE ε4 on the degenerative process was extensively spread (Figure 3C, APOE ε4 thickness over time in Aβ+ group); specifically, the effect was more pronounced among APOE ε4 carriers than among the overall Aβ+ sample (Figure 1B, time effect of Aβ+ status). In the Aβ− MCI group, APOE ε4 carriers experienced a greater decrease in cortical thickness within the default mode network (DMN) relative to noncarriers (Figure 3B, decreased cortical thickness in APOE ε4 carriers with Aβ−). Furthermore, a longitudinal analysis revealed the significant genetic effects of APOE ε4 on disease progression (Figure 3C, APOE ε4 thickness over time in Aβ− group). The topographic distribution of cortical atrophy × time interactions in APOE É›4 carriers with Aβ− MCI mostly corresponded to that of the overall Aβ− MCI sample (Figure 1B, Aβ− time effect). The effects of APOE ε4 were not detected in the controls. “ (page 8, line 203)
Point 14: Lines 190-191: It seems to me that in the sentence: “were topographically wider in which the areas related to short term memory and orientation (Figure 4)”, should be “were topographically wider than in areas related to short term memory and orientation (Figure 4)”
Response 14:
Sorry for the confusion. We rewrote the sentence in’ 2.7. Cortical–cognitive relationships in Aβ+ and Aβ− MCI groups’ section as follows:
“For the main effect, the regions correlated with MMSE or verbal fluency scores were spread across a wider area relative to those correlated with STM or orientation (Figure 4).” (page 9, line 222).

Round 2
Reviewer 2 Report
No further comments.